# The Complex Interplay between Endocannabinoid System and the Estrogen System in Central Nervous System and Periphery

**DOI:** 10.3390/ijms22020972

**Published:** 2021-01-19

**Authors:** Antonietta Santoro, Elena Mele, Marianna Marino, Andrea Viggiano, Stefania Lucia Nori, Rosaria Meccariello

**Affiliations:** 1Department of Medicine, Surgery and Dentistry “Scuola Medica Salernitana”, University of Salerno, 84081 Baronissi, Italy; mamarino@unisa.it (M.M.); aviggiano@unisa.it (A.V.); 2Department of Movement Sciences and Wellbeing, Parthenope University of Naples, 80133 Naples, Italy; elena.mele@collaboratore.uniparthenope.it; 3Department of Pharmacy, University of Salerno, 84084 Fisciano, Italy; snori@unisa.it

**Keywords:** endocannabinoid system, estrogens, central nervous system, reproduction, cancer

## Abstract

The endocannabinoid system (ECS) is a lipid cell signaling system involved in the physiology and homeostasis of the brain and peripheral tissues. Synaptic plasticity, neuroendocrine functions, reproduction, and immune response among others all require the activity of functional ECS, with the onset of disease in case of ECS impairment. Estrogens, classically considered as female steroid hormones, regulate growth, differentiation, and many other functions in a broad range of target tissues and both sexes through the activation of nuclear and membrane estrogen receptors (ERs), which leads to genomic and non-genomic cell responses. Since ECS function overlaps or integrates with many other cell signaling systems, this review aims at updating the knowledge about the possible crosstalk between ECS and estrogen system (ES) at both central and peripheral level, with focuses on the central nervous system, reproduction, and cancer.

## 1. Introduction

The endocannabinoid system (ECS) is a lipid cell signaling system required to maintain homeostasis. It is expressed from lower organisms to humans (except for insects) and this suggests a pivotal role of ECS for essential functions in animals [1,2,3]. The ECS system takes its name from the *Cannabis sativa* plant since some phytocannabinoids, including the main psychoactive principle Δ^9^—tetrahydrocannabinol (Δ^9^-THC), can mimic the effects of endocannabinoids by binding to their endogenous receptors [4]. From its discovery in 1964 [5], several studies tried to elucidate the pathophysiological role of ECS and the knowledge on this system increased enormously. However, studies have revealed a complex ECS function that overlaps or cooperates with many other cell signaling systems making it difficult to establish the exact role of the ECS in human cells, tissues, and organs. From this hand, the importance of the possible interactions between ECS and estrogen system (ES) emerged [6,7,8,9]. Therefore, this review aims to update the knowledge about the ECS and the ES cross-talk at both the central and peripheral levels.

## 2. The ECS System

The ECS is a ubiquitous cell signaling system that is involved in a wide range of physiological processes and pathological conditions including homeostasis, reproduction, cancer, inflammation, cardiovascular disease, and neurodegeneration [10,11,12,13]. It is constituted by lipid messengers named endogenous cannabinoids or endocannabinoids, the cannabinoid (CB) receptors, and the enzymes catalyzing their biosynthesis and degradation [14]. The known endocannabinoids are N-arachidonoylethanolamide (anandamide, AEA), 2-arachidonoylglycerol (2-AG), 2-arachidonyl glyceryl ether (noladine, 2-AGE), virodhamine (O-arachidonoyl ethanolamine), and N-arachidonoyl-dopamine (NADA).

The first identified endocannabinoid was AEA that was isolated in the pig brain in 1992 immediately after the discovery of the CB1 receptor [15]. It is an amide derivative of arachidonic acid but conversely to many other neurotransmitters that are stored in synaptic vesicles, AEA (and 2-AG) are produced “on-demand” by neurons from the cleavage of membrane phospholipid precursors, following the depolarization of cell membrane and intracellular increase in calcium levels (Ca^2+^) [4]. However, recently an adiposomic localization of AEA has been reported in different cell types highlighting the possibility of this endocannabinoid to be stored in internal organelles [16]. Once produced, AEA or 2-AG can be released from the pre- or post-synaptic membrane, in the intersynaptic space; then, they travel in a retrograde direction to bind CB receptors on presynaptic terminals [4]. Activation of CB receptors involves the inhibition of the activity adenylate cyclase, with lower production of the second cAMP messenger, ultimately leading to a hyperpolarization of the cell membranes and activation or inhibition of specific ion channels and signaling pathways that are capable to change intracellular metabolism. Thereafter, the removal of AEA and 2-AG from the presynaptic space takes place by a selective reuptake process mediated by a membrane transporter or by passive diffusion across the membrane. Once inside the cell, AEA and 2-AG are rapidly hydrolyzed by the fatty acid amide hydrolase (FAAH) and monoacylglycerol lipase (MAGL) enzymes with their consequent deactivation (see References [4,16] for recent reviews). Figure 1 summarizes the retrograde action of endocannabinoids and their uptake and hydrolysis in neurons.

### CB Receptors Expression in the Central Nervous System (CNS) and Peripheral Organs

Endocannabinoids exert their functions essentially by binding to CB1 and CB2 receptors. However, the evidence pointed to the existence of other receptors such as transient receptor potential cation channel subfamily V member 1 (TRPV1), the “orphan” receptors coupled to the G protein GPR55 and GPR119, and peroxisome proliferator-activated receptors (PPARs) that are capable of binding both centrally and peripherally endocannabinoids or endocannabinoid-like compounds [4,16]. The CB1 and CB2 receptors are encoded by *CNR1* and *CNR2* genes, respectively, and belong to the family of G protein-coupled receptors (GPCR) superfamily. They are seven transmembrane domain receptors bearing both extracellular amino terminal and carbonyl terminal intracellular domains [17]. Of note, CB1 and CB2 receptors share a significant homology however they differ in their function and specificity [18]. The CB1 receptor is most abundant in the CNS and is widely expressed in several brain areas with the highest concentration in regions associated with cognition, movement, and emotional behavior like the amygdala, hippocampus, septum, brain cortex, globus pallidus, substantia nigra, cerebellum, and putamen [19]. The CB1 receptor is localized preferentially in presynaptic glutamatergic and γ-aminobutyric (GABA) acid axon terminals; however, in the hippocampus, the CB1 receptor is located mainly in GABA-ergic, inhibitory interneurons and is present at lower concentrations in the hippocampal glutamatergic axon terminals [20]. Additionally, the CB1 receptor is present in the periphery both on sensory nerve fibers, the autonomic nervous system [18,21] and, among others, in breast, colon, and testis where it modulates cell proliferation, cell cycle progression, and cell differentiation [6,22,23,24].

The CB2 receptor has been found mainly outside the CNS and is particularly associated with immune tissues and organs, such as the spleen, the thymus, tonsils, and bone marrow, and is expressed in circulating immune cell populations in which it exhibits immunomodulatory functions [18]. However, in recent years CB2 receptors mRNA expression has been found at low concentrations in CNS neurons of the brainstem, cerebellum, and cortex [25,26] as well as in the internal and the external segments of the globus pallidus of the non-human primates [27], in human substantia nigra [28,29], in hippocampal glutamatergic neurons, [30] and dopaminergic neurons of the ventral tegmental area [31]. These findings suggest an important role of the CB2 receptor in the brain and have promoted investigations on the CB2 receptor in neural functions such as pain, movement, memory, and learning. On the other hand, upregulation of the CB2 receptor has been linked to various insults, such as chronic pain, neuroinflammation, and stroke [32,33]. Morgan et al., 2009 [34] have documented the presence of a functional expression of the CB2 receptor at the synapse level in the medial entorhinal area of the rat. In fact, they found that the pharmacological blockade of the CB1 receptor was not able to inhibit the effects induced by 2-AG on neurons, whereas CB2 receptor agonist JWH-133 suppressed 2-AG—induced inhibition of GABAergic neurons and this effect could be reversed by the CB2 receptor inverse agonist AM-630 suggesting a role for this receptor in neurotransmission [34]. According to the CB2 receptor role in the immune system, its expression has been found also in microglia, the only CNS cell type deriving from hematopoietic stem cells [35]. The role of the CB2 receptor in microglia is not fully understood, however, several studies evidenced a CB2 receptor up-regulation in activated microglia following pathological conditions such as inflammation, Alzheimer’s disease, and dementia [36,37,38,39]. Due to the complex function played by microglia either in neuronal proliferation and differentiation during CNS development or in neuronal functions and senescence in adults [35], these findings highlight the pivotal role that this receptor holds within the brain.

## 3. The ES

Estrogens are steroid hormones regulating growth, differentiation, and many other functions in a broad range of target tissues. Classically, the biological effects of estrogens are mediated through their interaction with estrogen receptor alpha (ERα) and beta (ERβ), which are members of the large superfamily of nuclear receptors. The ER dimers move from the cytoplasm into the nucleus and bind specific estrogen response elements (EREs) on the promoter region of target genes giving rise to defined effects “genomic” [40]. However, about one-third of the genes in humans that are regulated by ERs do not contain ERE-like sequences [41]. In fact, ERs can regulate gene expression also by modulating the function of other classes of transcription factors such as AP-1 through protein-protein interactions in the nucleus and this represents a typical ERE-independent genomic action [42]. Moreover, estrogens can exert rapid effects independently from the activation of RNA and protein synthesis. These actions are known as “nongenomic” effects and are probably mediated through plasma membrane-associated forms of ERs mERα and mERβ, the G protein-coupled receptor GPER1/GPER30, and ER-X [43]. mERα and mERβ derive from the same transcripts of ERα and ERβ and have been found in several peripheral and central tissues where they are responsible for rapid estrogen signaling [44,45]. Conversely, GPER30 is a transmembrane G protein-coupled ER-localized both in the plasma membrane and endoplasmic reticulum of several brain areas including the cortex, hippocampus, and striatum, and also expressed in liver, male and female reproductive tract, and in vascular smooth muscle cells [44]. Its activation protects against the neurotoxicity induced by N-methyl-d-aspartate (NMDA) exposure in cortical neurons and glutamate injuries in the retina [43]. Intriguingly, ER-X has been found in mice brain and uterus; it can interact with both 17α estradiol and 17β estradiol, even though its preferred ligand is 17α estradiol. ER-X expression is developmentally regulated probably having an important but yet undiscovered role in both the adult hippocampus and neocortex as well as in the uterus [46]. Indeed, in the brain and the uterus, ER-X has been found during the first postnatal month while it becomes scarcely detectable in the adult, suggesting that it could be implicated either in neuronal cortex differentiation or in prenatal sexual development [46]. The actions of plasma membrane ERs are frequently associated with the activation of various protein kinase cascades, however, ER-X stimulation can activate the MAPK/ERK pathway whereas its co-genre receptor ERα seems to be inhibitory [46,47]. The most potent estrogen in humans is 17β estradiol, but other estrogens such as 17α estradiol estrone and estriol are also present at lower concentrations. The immediate precursor of 17β estradiol is testosterone, synthesized by aromatizing ring A in a three-step reaction catalyzed by the aromatase enzyme complex. Circulating estrogens are produced by the adrenal glands, the gonads, and the placenta can affect the physiological processes of many peripheral target tissues. Furthermore, estrogens can readily cross the blood-brain barrier (BBB) and reach the CNS [48]. The CNS can also synthesize steroids which are involved in neurogenesis and synapses plasticity therefore affecting memory and learning [49,50]. Interestingly, the steroids, dehydroepiandrosterone (DHEA) and its sulfate ester (DHEAS), pregnenolone, and allopregnanolone have been found in the brain at concentrations higher than in the serum [48,51,52]. However, it is believed that neurosteroids and circulating steroids have similar effects in the CNS [53].

### 3.1. Estrogens in the CNS and Reproductive Tract

There is a strong consistency for an anatomical specific expression of ERs in the brain: it is clear that across species, ERα, ERβ, and GPER are widely distributed in brain regions that are not necessarily associated with reproductive functions. Gonadal hormones affect the nervous system in ways that extend beyond their capacity to regulate gonadotropin and prolactin (PRL) production. Estrogens and androgens have been reported to affect verbal fluency, performance on spatial tasks, and to affect the coordination of movement in animals [54,55,56,57]. Immunohistochemical staining reveals the presence of steroidogenic proteins including P450(17α) and P450 aromatase (also known as P450arom or Cyp19) and steroidogenic acute regulatory protein (StAR) in pyramidal neurons and granule cells of the Dentate gyrus (DG) of adult [58,59,60,61,62,63] and developmental hippocampus [64,65,66,67]. Interestingly, the synthesis of sex steroids by cytochrome P450(17α) and P450arom occurs also at the synaptic level suggesting that pyramidal neurons and granule neurons are equipped with complete steroidogenic systems which catalyze the conversion of cholesterol to pregnenolone (PREG), DHEA, and estradiol [61,67,68,69]. The synaptic synthesis of sex steroids (synaptocrine mechanism) proceeds through the first step of glutamate release from the presynapse inducing a Ca^2+^ influx through the NMDA receptors. The Ca^2+^ influx drives StAR to transport cholesterol into the mitochondria, where the cytochrome P450 monooxygenase (P450scc) converts cholesterol to PREG. This synaptocrine mechanism highlights another mechanism of estrogen and androgen synthesis in addition to the classical microsomal synthesis of sex steroids in peripheral tissues (References [70,71,72], for reviews). At the periphery, the activity of Cyp19, the enzyme that locally catalyzes the demethylation of androgens’ carbon 19, producing phenolic 18-carbon estrogens is found in several human tissues and cells, including ovarian granulosa cells, the placental syncytiotrophoblast, testis, bone, skin fibroblasts and adipose tissue [73,74].

In the gonads, the production of sex steroids is centrally directed by the hypothalamic gonadotropin-releasing hormone (GnRH) (see Section 5.1 for further details). In the ovary, the 2-cell 2-gonadotophin process is responsible for estradiol production. In fact, during follicle maturation theca cells which are responsive to pituitary luteinizing hormone (LH), produce androgens (predominantly androstenedione and testosterone) that diffuse along the basal lamina of the follicle to reach the Follicle Stimulating Hormone (FSH)-responsive granulosa cells possessing Cyp19 activity [75].

Classically considered as a female hormone, demonstrations of estrogen synthesis in the testis and high concentrations of 17β-estradiol in rete testis fluid together with the extensive-expression of ERs within the male reproductive tract from the neonatal period to adulthood have been provided [74]. Key roles in male reproduction with involvement in gametogenesis, sperm maturation, and the developmental process of the male reproductive tract have been reported [74,76]. Despite these observations, the gonadal source of estrogen in testis is still debated. In fact, interstitial Leydig cells are the archetype cell previously accepted as the sole estrogen source in adult testis, but several authors suggest Cyp19 activity also in the germinal epithelium [74].

### 3.2. Different Functions of ES in Male and Female CNS

There is mounting evidence that estrogens may have opposite effects in male and female brains principally due to differences in brain organization. The developing brain expresses high levels of ERs and estradiol mediated cellular end-points including morphometry of neurons and astrocytes, the promotion or the inhibition of apoptosis and synaptogenesis in different brain regions [77]. For example, estradiol has opposite effects on several brain areas in males, being simultaneously capable of promoting either the apoptosis in a critical region for female ovulation like the anteroventral periventricular (AVPV) nucleus of the hypothalamus and neurite outgrowth and cell survival in the bed nucleus of the stria terminalis and medial amygdala [78].

Sexual differentiation of the brain requires the activity of Cyp19 and ERα signaling within the hypothalamus to drive the maturation of GnRH neurons and the regulation of sexual behavior through epigenetic mechanisms [73]; consistently, epigenetic modulation of the process by exogenous factors like the exposure to chemicals that exhibit endocrine-disrupting activity has been reported [79]. Therefore, transient exposure to estradiol during critical developmental periods may induce epigenetic changes in the chromatin state of gene regulatory elements within different brain areas, leading to sex-specific differences in both gene expression and transcriptional response to later environmental cues [79].

Sex-specific differences in brain organization are critical for the development of therapies for the treatment of brain disorders that differentially affect men and women [70]. In clinical cases with inactivating mutations in the *CYP19* gene, several physiological disturbances have been identified in men, including skeletal, metabolic, and reproductive impairments [80,81,82]. Studies in aromatase knockout (ArKO) mice generally recapitulate these sequelae in peripheral physiology and, in addition, reveal important functions of estrogens in both male and female brains, thereby highlighting the ubiquitous distribution and function of the aromatase enzyme in peripheral and central tissues.

In the adult brain, the highest levels of aromatase activity have found in the hypothalamus of all species studied, especially within the preoptic area (POA) and ventromedial nucleus (VMN), where the enzyme is regulated by gonadal steroids and found at higher levels in males than in females [83]. Apart from the hypothalamus, significant levels of aromatase are also found in other brain regions, including the amygdala, hippocampus, midbrain, and cortical regions in rodents, nonhuman primates, and humans, where its expression is steroid-independent and not significantly different in males and females. It is also known that central actions of estrogens may occur via ER-independent mechanisms, which would persist in ER-null mice. Therefore, since the ArKO mice lack the classic ability to synthesize both peripheral and central estrogens, they have provided insights into the CNS roles of estrogens. These include the intriguing observations that in the absence of estrogen synthesis, apoptosis of dopaminergic neurons occurs spontaneously in the adult male, not female, hypothalamus, whereas apoptosis of pyramidal neurons in the frontal cortex occurs spontaneously in the adult female, but not male, brain. This highlights a notable sex dimorphism in the requirement and/or the ability of estrogen to maintain specific neuronal populations in different brain regions. Although the underlying mechanisms and the functional consequences of these morphological changes are unknown, sex- and age-specific behavioral deficits have been identified in ArKO mice [84,85] and support the concept that estrogens play a sexually dimorphic role in the CNS.

Recent data suggest that sexually differentiated intracellular signaling pathways may represent a further mechanism underlying sex-specific responses to estradiol in the brain. In the POA and the VMN, the numbers of cells expressing phosphorylated (activated) cAMP response element-binding protein (CREB) were significantly increased within minutes of treating gonadectomized female mice with estradiol, but this effect was not seen in males [86]. As phosphorylation of CREB is a necessary step in the estrogen-dependent generation of new dendritic spines (at least in cultured hippocampal neurons) [87], sex differences in this signaling mechanism may account for sex-specific effects of estradiol on spine density in the VMN.

Another difference is about synapse density. Estrogens regulate synapse density in the adult rat hypothalamic ventromedial nucleus that differs between males and females [88,89,90]. This discovery led to the finding that the ovarian cycle regulates cyclic synaptogenesis on excitatory spines in hippocampal CA1 pyramidal neurons in female but not in male rats [91,92]. Synaptogenesis is cyclic, and fluctuations in synapse density occur throughout the estrous cycle of the female rat. Male rats show much less estrogen-induced synapse formation unless they are treated at birth with an aromatase inhibitor [70].

## 4. ECS and ES Interactions in the CNS

The interplay between ECS and ES in the CNS has been little investigated. Few studies demonstrated that the modulation of ECS is involved in the mechanisms of neuroprotection afforded by acute administration of pharmacological doses of estrogen in male rats [93]. Authors, aiming to evaluate the effect of middle cerebral artery occlusion (MCAo)-induced brain insult on AEA regional brain level and receptor binding, found that an intraperitoneal dose of 17β estradiol (0.20 mg/kg) could reverse the endogenous increase of AEA in ischemic striatum. Moreover, brain ischemia did not alter CB receptor expression whereas 17β estradiol pre-treatment resulted in a 45% reduction of CB1 receptor binding in the striatum. The treatment with 17β estradiol also modulated N-acyl-phosphatidylethanolamine-hydrolyzing phospholipase D (NAPE-PLD) and FAAH expression and activity, suggesting that the hormone could have a neuroprotective effect by reducing endogenous cannabinoids levels and their ability to induce CB1 receptor-mediated response [93]. FAAH modulation by estrogens was corroborated in studies of Hill and co-workers (2007) [94] who analyzed the mechanisms by which estrogens elicit anxiolytic and antidepressant-like effects. They investigated the ability of estrogens to modulate the ECS in ovariectomized rats. Results showed that both the pharmacological inhibition of CB1 receptor and the administration of the FAAH inhibitor URB597 were able to reverse the significant increase in open arm entries in the elevated plus-maze (EPM) test and time spent in the center quadrant of the open field test (OFT) following 17β estradiol treatment [94]. These findings demonstrated that estrogen may produce changes in emotional behavior by modulating ECS activity mainly through FAAH inhibition. Finally, some studies demonstrated that pregnenolone, the inactive precursor of all steroid hormones, acting as a signaling-specific inhibitor of the CB1 receptor, reduces the toxicity of Δ^9^-THC [95], proposing that ECS could be a target for the development of neurosteroidogenic drugs [96]. The above data suggest that estrogens can modulate directly ECS in the CNS, however, studies are still few and inconclusive. In view of the presence of the CB2 receptor and the different distribution and concentration of the CB receptors in the brain, it would be desirable that studies focus on both CB receptors expression and activity and better investigate the role of endocannabinoids degrading enzymes following estrogen treatments. This could improve the understanding of neurodegeneration and neuroinflammation processes opening new perspectives in drug development.

## 5. ECS, Estrogens, and Reproduction

Reproduction, the process that leads to the production of gametes, is strictly controlled by the activity of the hypothalamic gonadotropin-releasing hormone (GnRH), a decapeptide released in a pulsatile manner within the hypophyseal portal system to stimulate the adenohypophysis to release in the main circulation the pituitary gonadotropins (Luteinizing hormone (LH) and Follicle Stimulating Hormone (FSH)). FSH and LH reach the gonads which in turn produces sex steroids to sustain spermatogenesis in males, follicle development, oocyte maturation, and endometrium receptivity to fertilized oocytes in females [97,98]. Notably, in males testosterone exerts negative feedback on GnRH secretion; in females, estradiol exerts homeostatic, negative feedback on GnRH secretion, but at the end of the follicular phase, the rise in estradiol levels causes the switch into a positive feedback that sustains the GnRH peak and LH surge responsible for ovulation [99] (Figure 2).

However, within the male and female reproductive system, nuclear and membrane estradiol signaling exerts fundamental activities resulting critical for proliferative events, but also sperm maturation and transport in males [76], ovarian granulosa cell differentiation, follicle and oocyte growth and development, ovulation, endometrial growth and uterus lining in females [100].

In addition, several centrally and peripherally produced modulators, regulate the activity of the hypothalamus-pituitary gonad (HPG) axis and steroid production in response to exogenous environmental cues like diet, stress, or drug addiction, with consequences on gamete quality, fertility rate, and pregnancy [11,76,79,101,102,103,104].

ECS both centrally and locally affects the activity of the HPG axis in both sexes. Receptors and metabolic enzymes have been detected within the hypothalamus, anterior pituitary, testis, ovary, and reproductive tract [105,106], and endocannabinoids detected within the gonads, in the reproductive tract and reproductive fluids [105,107]. ECS involvement in the modulation of sex steroid biosynthesis, gamete quality, fertilizing ability, implantation, placentation, and embryo development have been reported [105,106,108,109,110,111,112,113]. As a consequence, the genetic impairment of the system and its epigenetic modulation by lifestyle may have a deep impact on reproduction and fertility in both sexes with possible trans-generational effects ([104] and References inside).

In general, data from in vivo, in vitro, and clinical studies, have reported that cannabinoids in marijuana (i.e., Δ^9^-THC) can reduce fertility by disrupting the hypothalamic release of GnRH, leading to reduced levels of LH and sex steroids in both sexes, anovulatory cycles and depressed ovarian follicular maturation in females, impaired spermatogenesis, and sperm functions (e.g., motility, capacitation, and the acrosome reaction) in males [105,108,109,112]. Implantation failure, inhibition of early embryo development, increased incidence of miscarriage, preterm birth, prematurity, and low fetal birth weight are the main consequences of marijuana in animal models and marijuana smoker women [104,113]. Interestingly, steroids influence the deleterious effects of marijuana, with estrogen generally increases and progesterone decreasing sensitivity to marijuana [109].

Δ^9^-THC, the main bioactive constituent of marijuana has been recognized to possess anti-estrogenic activity due to its ability to up-regulate the nuclear ERβ, disrupting estrogen-ER*α*-signaling, and inhibiting the expression of estradiol/ER*α*-regulated genes in cell lines [114]. Similarly, the CB1 receptor antagonist AM251 up-regulates the expression rate and signaling of epidermal growth factor receptor (EGFR) via a CB1 receptor-independent pathway involving the destabilization of ER related α (ERRα) in PANC-1 and HCT116 cancer cell lines [115].

As for marijuana/Δ^9^-THC, the manipulation of ECS or its genetic impairment have deep consequences on reproduction in animal models, with poor gamete quality, (sub)fertility, gestational and developmental troubles in *Cnr1*^−/−^, *Cnr2*^−/−^ and *Faah*^−/−^ mice as reviewed elsewhere [8,108,116,117,118,119,120,121,122].

The neuroendocrine axis and the consequences of gametogenesis and gamete quality have been deeply analyzed in male *Cnr1*^−/−^ KO mice [8]. Mutant mice display normal pituitary LH content but low serum LH concentration, low testosterone secretion from the testis, and low circulating levels of testosterone and estradiol. The expression levels of *GnRHR* and *FSH β* subunit were respectively increased and decreased in the pituitary; in the testis low expression of *FSH receptor* (*FSHR*) and *Cyp19* have been reported; a low number of adult Leydig cells [23] expressing low *Cyp19* and normal *3β*-*hydroxysteroid dehydrogenase* (*3β*-*HSD*) mRNA levels complete this intricate neuroendocrine speech [8]. In parallel, precocious acquisition of sperm motility in the epididymis [123] and poor chromatin quality indices (e.g., high histone content, uncondensed chromatin, DNA damage, elongated nuclear size, impaired disulfide bonds formation) [124,125,126,127] have been reported, with estradiol administration capable of restoring histone displacement in spermatids [124,126].

In the female reproductive tract, the expression of some components of the ECS is directly modulated by estrogen. Maia and co-workers recently investigated the estradiol benzoate (EB) and tamoxifen dependent modulation of ECS in ovariectomized rats revealing that the expression of CB1 and CB2 receptors, cyclooxygenase-2 (COX-2), and AEA-metabolic enzymes (i.e., NAPE-PLD, FAAH) were higher in the uterus following EB administration; in parallel, the levels of AEA were higher in plasma, but not in the uterus whereas the production of prostaglandin (PG)E_2_—one of the major products of COX-2- were higher in the uterus [128].

In human granulosa cell line KGN the expression of a complete ECS comprising receptors, biosynthetic and hydrolyzing enzymes (i.e., the CB1 receptor, the CB2 receptor, DAGL, FAAH, GPR55, MAGL, NAPE-PLD, and TRPV1) occurred independently from FSH administration [129] and manipulation of CBs affected *Cyp19* expression rate with consequences on estradiol but not progesterone biosynthesis. In addition, receptor-dependent changes in the expression levels of microRNA (miRNA) specifically involved in the activity of granulosa cells like *miR*-*23a*, *miR*-*24*, *miR*-*181a*, and *miR*-*320a* have been observed [129], thus revealing the requirement of a functional ECS for appropriate estradiol synthesis by the ovary.

However, the modulation of endocannabinoid tone is always critical in both female and male reproduction [105,108]. For instance, reduced fertility in the *Faah*^−/−^ mice is due to persistent or elevated endocannabinoid tone/signaling in testis, epididymis, oviduct/uterus, a situation that impairs the physiological environment critical for successful gametogenesis, sperm maturation, normal development of embryos and oviductal transport and implantation [119,122]. Interestingly, in mammalian and non-mammalian vertebrates FAAH, the main endocannabinoid hydrolyzing enzyme, is expressed in an estradiol dependent manner [130,131,132]. Accordingly, in mouse primary Sertoli cells, estradiol via the nuclear ERβ binds two EREs within the *FAAH* promoter and activates *FAAH* transcription [132]. An epigenetic mechanism involving the histone demethylase LSD1, that acts decreasing methylation of both DNA at CpG site and histone H3 at lysine 9, has been reported [128]. Interestingly, the stimulation of *FAAH* expression causes the increase in FAAH protein activity, decreasing AEA levels, and lastly protecting Sertoli cells from AEA-induced apoptosis [133].

Taken together, a deep interplay links ECS and estrogens to gain successful reproduction. Therefore, in the next sections, we summarize the recent findings related to the ECS-dependent modulation of GnRH and the interplay with estrogens.

### 5.1. The ECS-Dependent Modulation of GnRH and the Interplay with Estrogens

The aforementioned ECS effects on reproduction in both sexes are the result of combined centrally and locally mediated effects. Centrally, the effects of the ECS on the secretion of GnRH and the interplay between ECS and estrogens have been documented with direct and mediated effects [108,110,111], but molecular mechanisms are not clearly understood.

Notably, the CB1 receptor is largely expressed within the hypothalamus in neuronal populations, and ex vivo, in vitro, and in vivo endocannabinoids and phytocannabinoids negatively affects GnRH secretion [108]. At the molecular level, retrograde endocannabinoid signaling via CB1 receptor affects the neuronal networks and the release of neuromodulators involved in the secretion of GnRH like glutamate and GABA. In particular, 2-AG via CB1 receptor reduces the GABAergic synaptic transmission negatively affecting the hypothalamic release of GnRH in male rodents [134]. In addition, direct effects have been also reported in mammals [134] and in the diencephalon of non-mammalian vertebrates, being CB1 receptor co-localized on a subset of GnRH secreting neurons [134,135]. Interestingly, endocannabinoid involvement in the modulation of negative sex steroid feedback has been recently reported throughout the metestrus phase of the ovarian cycle in female mice [136] and, the requirement of ERβ and 2-AG/CB1 receptor, but not ERα and GPR30, has been demonstrated in gonad-intact transgenic female mice by means of electrophysiological experiments [136].

The modulation of the HPG axis by environmental cues such as stressors or diet reveals the interplay between different neuronal networks and signaling pathways including the ECS [11]. Reproduction strongly depends on energy homeostasis, especially in females [137] and endocannabinoids are well-known regulators of food intake and gut-brain communications [11,138,139,140]. The possible interplay between estrogen and cannabinoid signaling has been evaluated in ovariectomized female guinea pigs through the use of the CB1 receptor agonists WIN 55,212-2, EB, and the specific agonists of ERα, ERβ, mERs, and GPR30 (PPT, DPN, STX, and G-1 respectively). Data revealed that in the arcuate nucleus (ARC) of females, estrogens negatively modulate cannabinoid-induced changes in energy balance via genomic and nongenomic mechanisms, with the downstream involvement of protein kinase A and C [141].

In such a context kisspeptins, the cleavage product of the Kiss1 precursor, represents the main gatekeepers of the HPG axis being the major upstream modulators of GnRH secreting neurons at puberty onset, a mediator of sex-steroid positive/negative feedback mechanisms, peripheral modulators of gonad physiology in both sexes and biosensors of metabolic status therefore subjected to epigenetic modulation [76,98,102,142,143]. Currently, the possible link between kisspeptin system and ECS has been investigated in mammals following stress, revealing that in male rats 1 h immobilization stress resulted in a decrease in the serum LH concentration and a drop in *kiss1* expression in the medial POA, and to some extent the ARC, key hypothalamic regions that regulate GnRH release. Central administration of the CB1 receptor antagonist AM251 completely reversed this effect confirming CB1 receptor involvement [144].

Recently, a molecular mechanism involving both the kisspeptin system and FAAH has been suggested in the anuran amphibian, the frog *Pelophylax esculentus*, a seasonal breeder with a complete, well-characterized ECS [145,146] involved in the central and local control of reproduction with effects on steroid biosynthesis and sperm maturation [131,135,147,148,149,150,151]. As for mammals, in this animal model AEA via CB1 receptor centrally impairs the hypothalamic activity of the GnRH system [135], which in the frog comprises two molecular forms (GnRH1 and GnRH2) as ligands and three receptors [148], thus centrally modulating steroid biosynthesis [150]. As reported by Ciaramella et al., in vivo AEA treatment had different outcomes on the HPG axis in frog: (i) decreased the expression levels of Kiss1 and Kiss1R protein and *GnRHs* and *GnRHRs* mRNA in brain and testis; (ii) impaired the expression of key steroidogenesis enzymes (i.e., *Cyp17* and *3ß*-*HSD*) causing a drop in intratesticular testosterone level; (iii) increased the expression of Cyp19, causing the increase of intratesticular estradiol levels, and the estradiol dependent expression of FAAH, [131]. Interestingly, the kisspeptin system not only is the major upstream modulator of GnRH, but it is also involved in sex steroid feedback mechanisms [98], and as both ECS [149] and GnRH system [152], is an intratesticular modulator of spermatogenesis [142,153,154,155] with functions related to Leydig cells activity, spermatogenesis progression [142,153,154] and a dose-dependent suggestive role in the balance of estradiol and testosterone levels [155]. To discriminate between centrally mediated and direct intragonadic effects, ex vivo incubations of the testis with AEA ± rimonabant (SR141716), the antagonist/inverse agonist of CB1 receptor were carried out, revealing direct CB1-mediated effects on FAAH, Cyp19, and GnRH system [131,149]. Since the AEA inductive effects on FAAH, the possible involvement of estradiol signaling were assayed incubating frog testis with ICI182-780—a potent ER inhibitor and therefore stimulating with AEA. The preliminary administration of ICI182-780 locked any positive effects of AEA on FAAH expression, providing evidence of combined centrally-mediated, and intratesticular route in the control of sex steroid biosynthesis. Centrally, AEA negatively affects the hypothalamic kisspeptin system to switch off the HPG axis, thus causing testosterone decrease; locally, testosterone is further converted into estradiol, which in turn induces FAAH, indicating the existence of a fine regulatory loop in the modulation of AEA tone (Figure 3).

## 6. ES and ECS Modulation by Estrogens-Like Substances Affecting CNS and Gonads

Several natural and synthetic pseudoestrogens including compounds like phytoestrogens, pesticides, parabens, and plasticizers among the others, exhibit estrogenic or antiestrogenic properties and thus are capable of interfering with the physiological signaling of steroids, with consequences on endocrine functions at the central and peripheral level and long-time pleiotropic effects on exposed organisms [156,157,158]. The majority of harmful effects of endocrine disruptor chemicals (EDCs) have been attributed to molecular mechanisms mediated by nuclear receptors, however recent data support the hypothesis that EDCs can act also through other receptors including the activity of pregnane X receptor (PXR), ERRγ, thyroid receptor (TRs), retinoid X receptors (RXRs), PPARα, or PPARγ [159].

Substances like Bisphenol A (BPA, 2,2-bis (4-hydroxyphenyl) propane), dibutyl-phthalate (DBP), bis-(2-ethylhexyl)-phthalate (DEHP) or perfluorinated compounds (PFCs) are EDCs worldwide diffused as an environmental contaminant and are commonly used in the production of daily use goods including food and drink packaging [102,160,161,162]. In the last year, particular attention has been devoted to BPA, a plasticized largely used for the production of epoxy resins and polycarbonate plastics that is capable of interfering in the endocrine system with main outcomes on reproductive health and the possibility of trans-generation epigenetic inheritance in the offspring [76,79,102]. Currently, oral exposure by ingestion of contaminated foods and drinks represents the major exposure route [102,160], but this compound accumulates in biological tissues and its metabolites are excreted in biological fluids [160]. Furthermore, it crosses both the placental barrier and blood-brain barrier interfering in developmental processes [163,164]. In line with this, we have recently shown that rat offspring exposed to BPA, first via the placenta and during lactation and drinking water later, induced astrocytosis and DNA damage in the hippocampus by modulating the normal expression of ERα [165]. We also found that BPA altered the first round of spermatogenesis with impairment of blood-testis barrier, reactive oxygen species production, and DNA damage [166], strongly suggesting that this EDC can affect both CNS and gonad development and differentiation. As a consequence, gestational and neonatal phase, childhood and adolescence represent the main exposure windows critical for health [79,102].

Another class of EDCs widely diffused in the environment both as natural products derived from plants and bacteria or synthesized industrially are parabens [158]. They are used as preservatives to prolong the shelf life of cosmetics, personal care, and food and have been documented to interfere with both androgen and estrogen levels by reducing Cyp19a1 [167]. Indeed, studies revealed that higher urine levels of methylparaben (MeP) and propylparaben (PrP) reduced couple fecundity [168] and PrP influenced negatively follicle reserve and ovarian ageing in women [169]. Results on parabens effects in CNS are quite scanty but some authors reported memory and learning deficits in BuP-treated rats suggesting that BuP might induce neurodevelopmental disorders [170]. On the contrary, Lara-Valderrábano et al., (2017) [171] observed that PrP suppressed the epileptiform activity of CA1 pyramidal neurons in vitro supporting the beneficial effects of this paraben in the management of some CNS diseases. Interestingly, parabens have been shown to interact also with ECS inhibiting FAAH activity, activating CB1 receptor, and PPARγ with the consequence of promoting adipocyte differentiation [172]. 

Data from fish recently revealed changes in the expression rate of genes involved in the metabolism of endocannabinoids or in endocannabinoid signaling in both liver and brain following the dietary exposure to BPA, diethylene glycol dibenzoate (DEGB), diisononyl phthalate (DiNP), or nonylphenol (4NP) with altered levels of endocannabinoids, and endocannabinoid-like mediators or altered expression rate of ECS components [173,174,175,176]. Interestingly, DEGB acts as an agonist of PPARα to stimulate key lipolytic genes and simultaneously down-regulates the genes encoding for endocannabinoid metabolic enzyme in the liver of sea bream [175]. Based on the findings that EDCs can target PPARα and PPARγ, it is conceivable that EDCs could promote developmental, reproductive, and neurological diseases in humans by interfering with the physiological functions of ECS and ES and their complex interplay. In 1-year-old zebrafish environmental BPA exposure altered the intragonadal levels of endocannabinoids and increased the expression of liver vitellogenin in both sexes. Specific changes in the gonads were also observed. In fact, in males, BPA increased the gonadosomatic index (GSI) and reduced the testicular area normally reserved to spermatogonia; in females, it increased the percentage of vitellogenic oocytes in the ovaries [177].

Similarly, in the male gilthead sea bream (*Sparus aurata*) fed with low and high concentrations of BPA (4 and 4000 µg/kg body weight respectively) for 21 days during the reproductive season, high exposure dose increased GSI, impaired sperm motility, increased testosterone level, reduced 11-ketotestosterone levels in plasma with both low and high doses significantly decreasing the intratesticular levels of endocannabinoids (AEA, 2-AG) and endocannabinoid-like compounds [i.e., palmitoylethanolamide (PEA) and oleoylethanolamine (OEA)] [178] with a parallel increase in FAAH activity. At the molecular level, the expression rate of genes encoding for the CB1 receptor, the CB2 receptor, and TRPV1 proteins were dose-dependently affected but not those encoding for the enzymes NAPE-PLD, FAAH, and DAGLα [178].

As for BPA, in the same animal models also the plasticizer DiNP had a similar effect on endocannabinoid tone and FAAH activity [179], with consequences on gonad physiology and the production of qualitatively good gametes; at molecular levels, specific changes in the expression rate of ECS components have been reported.

Low BPA doses (i.e., 0.5 μM for 48 h) modulate the ECS components in exposed mouse primary Sertoli cells in parallel to an increase in inhibin B levels; in this system, the BPA dependent effects on inhibin B are further enhanced by the blockade of either CB2 or TRPV1 receptors [180] revealing the involvement of CB2- and TRPV1-dependent signal transduction in the BPA mediated effects.

Recently the effects of BPA on ECS have been evaluated in pregnant women, providing evidence of BPA interference in ECS signaling and hypothesizing that BPA may induce adverse pregnancy outcomes modulating the activity of the ECS [181]. In this study liquid chromatography-mass spectrometry (LC-MS)-based plasma metabolomics was performed in pregnant women with known concentrations of free, conjugated, and total BPA. Positive correlations were observed between fatty acid amides and free and total BPA concentrations whereas OEA was positively correlated with conjugated BPA and lysophosphatidylethanolamine (LPE) with free BPA. Interestingly, in vitro study revealed that BPA caused a 15% inhibition of FAAH activity, therefore resulting in a competitive inhibitor capable of blocking FAAH catalytic residues [181]. Since FAAH activity is critical for regulating both the magnitude and duration of AEA signaling, this study has particular relevance in that BPA exposure may lead to a rise in circulating endocannabinoids, a well-known risk factor for miscarriage.

ECD exposure has been suggested to be a possible cause of the decline in human semen quality observed in the last few decades [182,183], with the possible paternal inheritance of epimutations in the offspring [184,185,186]. Therefore, the need to reduce chemicals in the environment to safeguard male fertility. Notably, marijuana abuse is considered an exposure to endocrine-disrupting factors mainly due to the estrogenic activity of the phenolic compounds contained in marijuana smoke condensate [187]. Nevertheless, the use of cannabinoids for recreational use interferes in the endogenous endocannabinoid tone with consequences on male fertility [105] and point out the epigenetic risk for the offspring [104]. Therefore, the relationships between sperm quality, endocannabinoid profiles in plasma/semen, xenobiotics (i.e., bisphenol A and S), and phytocannabinoids in urine have been recently evaluated in a cohort of 200 young Swiss men (age range 18–22) [188]. An inverse relationship between sperm motility and AEA concentrations in seminal fluid and OEA levels in blood serum has been found; conversely, PEA levels in semen were positively linked to sperm concentration, whereas in seminal fluid the levels of OEA and PEA—that exhibit antioxidants properties [189]—were associated with better sperm morphology. The presence of urinary THC-COOH (an inactive metabolite of Δ^9^-THC), used as an indicator of cannabis consumption, in *n* = 15 individuals has been linked to lower concentrations of endocannabinoids in semen and low circulating AEA in blood, suggesting a possible Δ^9^-THC-mediated negative feedback on AEA production and concentration; by contrast, no correlation was observed between the presence of urinary bisphenols and endocannabinoids [188].

Taken together, the world’s widespread distribution of EDCs points out a concrete risk for reproductive health and ECS may represent a target for disease, but also a biochemical marker for the toxicological screening of exposed subjects.

## 7. ECS and ES Interactions in the Periphery

### 7.1. Interactions in Blood Cells

Cardiovascular diseases are the leading cause of death in postmenopausal women in developed countries, suggesting that estrogen deficiency may play a causative role [190]. A positive effect of estrogen on plasma lipids and lipoproteins, endothelium-dependent vasodilation, and intimal hyperplasia has been documented [191] and is thought to contribute to the cardioprotective effects observed in women receiving estrogen replacement therapy. In addition, platelets have a well-established role in the pathogenesis of atherosclerosis and cardiovascular diseases [192]. The mechanisms by which estrogen influences coronary arteries and protects blood vessels against atherosclerotic development remain unclear. Yet, recent evidence suggests that the rapid effect of estrogen on vascular reactivity depends on the activation of surface receptors in endothelial cells [193], followed by endocannabinoid release in analogy with the nongenomic effect of glucocorticoids in the hypothalamus-pituitary-adrenal axis. In fact, in human umbilical vein endothelial cells (HUVECs) estrogen has been shown to activate the AEA membrane transporter (AMT) through calcium and nitric oxide (NO)-dependent mechanisms [194]. Estrogen stimulates also AEA synthesis via NAPE-PLD and inhibits nongenomically AEA degradation via FAAH, overall increasing the endogenous level of endothelial AEA. Therefore, released AEA can exert its manifold actions on the cardiovascular system, spanning from vasodilation to modulation of cell migration [195]. In addition, AEA was found to inhibit 5-hydroxytryptamine (5-HT) secretion from platelets, whereas estrogen did not [194], thus demonstrating that endocannabinoids can complement the biological activity of steroids.

Also, human T lymphocytes witness an intimate cross-talk between steroid hormones and ECS, with critical implications for human reproduction [196]. In circulating T cells, progesterone up-regulates FAAH, but not CB1 receptors, NAPE-PLD, or AMT, and reduces AEA content [197]. Of interest is the fact that progesterone exerts this effect genomically, i.e., by binding to its intracellular receptor (PR) and thus enhancing the level of the transcription factor Ikaros (Ik); this, in turn, increases FAAH gene expression by binding to a specie-specific sequence in the promoter region [197].

### 7.2. Interactions of ECS and ES in Cancer

The ECS is present in both central and peripheral organs and several findings suggest that endocannabinoids levels, ECS degrading enzyme activity as well as CB1 and CB2 receptor expressions are altered in sex-hormone dependent malignant tissues [198] as summarized in Table 1.

In hepatocellular carcinoma, CB1 and CB2 receptor expressions have been shown higher in well-differentiated cancers compared to poorly differentiated ones [199]. Moreover, Caffarell et al., (2010) [205] reported that highly aggressive ErbB2-positive breast cancer showed an upregulation of the CB2 receptor which correlates with ErbB2 expression. In contrast, CB1 receptor immunoreactivity was detected only in 14% of the tumors and no correlation was found with ErbB2. In normal mammary tissues, no significant CB1 or CB2 receptor immunoreactivity was detected suggesting that the overexpression of both CB receptors could be implicated in the development of this type of cancer [205]. However, not only in the canonical estrogen-dependent cancers but also in colon cancer—where the estrogen system has been suggested to play a crucial role in neoplastic transformation and cell proliferation [212,213]—is affected by ECS alterations. Indeed, some authors showed a CB1 receptor mRNA upregulation in normal colon tissues compared to colon cancer samples probably due to the hypermethylation of the *CNR1* gene encoding for the CB1 receptor [209]. On the other hand, cannabinoids (particularly Δ^9^-THC and cannabidiol, CBD) and endocannabinoids like AEA inhibited cancer cell growth and metastasis formation both in vitro and in vivo reviewed in [214,215,216]. Some evidence also indicated that AEA and 2-AG reduced colon cancer cell proliferation activating the CB1 receptor [215]. Similar results were obtained by the treatment with high pharmacological doses of the CB1 receptor antagonist/inverse agonist SR141716 that was able to inhibit cell proliferation in breast and colon cancer in the last case activating rather than suppressing the CB1 receptor [210,217]. From this point of view, the evidence seems to indicate a wide interplay between estrogen hormones and ECS [7]. The first evidence for a control of the ECS on estrogen system was provided by De Petrocellis et al., (1998) [206] and Melck et al., (2000) [200] who showed that AEA treatment was able to inhibit the proliferation of prolactin (PRL)-responsive human breast cancer cells and this effect was mediated by the down-regulation of the PRL receptor. Similar results were obtained by the same group in PRL- responsive human prostate DU-145 cancer cells in which the PRL-induced cell growth was potently inhibited by both AEA and 2-AG [200]. Furthermore, Bifulco’s group [22,207] found that CB1 receptor stimulation by AEA reduced cell proliferation in both ER-positive and ER-negative breast cancer cells being effective in inhibiting adhesion, migration as well as the epithelial-mesenchymal transition during cancer progression [188]. The intriguing effect of ECS stimulation as a cancer suppressor in estrogen-dependent tumors was furtherly investigated starting from the observation that 17β estradiol was capable to induce CB1 receptor expression in colon cancer cells [218]. It was shown that the overexpression of the CB1 receptor by 17β estradiol in DLD1 and SW620 cells was mediated directly through the activation of the *CNR1* gene promoter [6]. The same authors also found that increased levels of AEA either by exogenous administration or by using the FAAH inhibitor URB597 induced ERβ transcription and expression suggesting that the cross-talk between ECS and ES could control colon cancer cell proliferation [213]. Recently, it has been demonstrated the presence of an active ECS system also in endometrial cancer [219], in fact endogenous cannabinoids and CBD induced a significant reduction in cell viability in both Ishikawa and Hec50co cells by the activation of the apoptosis pathway, instead, Δ^9^-THC did not cause any effect [208]. CB1 receptor expression upregulation has been observed in prostate cancer tissues [204] and elevated levels of this receptor are associated with cancer severity and outcome [203]. Accordingly, AEA has been found to exhibit antiproliferative effects in LNCaP, DU145, and PC3 prostate cancer cells by acting through cannabinoid CB1 receptor and this led to an inhibition of the EGF-stimulated growth of these cells [201]. However recent studies documented that the cannabinoids CBD was able to reduce androgen receptor (AR)-positive (LNCaP and 22RV1) and AR-negative (DU-145 and PC-3) cells in a CB receptors independent manner [202], suggesting that CB1 receptor could be the main cannabinoid receptor that regulates the complex carcinogenic steps leading to prostate cancer.

Altogether these results underline a protective role for ECS in steroid-hormone-dependent cancers highlighting that the ES can regulate the ECS during tumor development and progression.

Although ECS is abundantly in the CNS and the role of estrogen is documented in brain development and in several physiological processes of the CNS like memory and learning [79,104], until now there is no direct evidence of a possible interplay between ECS and ES in brain cancer. Recent studies found that astrocytomas, the most prevalent type of glioma could be influenced by sex steroid hormones (reviewed in [220]) and high expression of the orphan nuclear ERRα is a marker of poor prognosis in patients with glioma [221]. The Authors demonstrated that ERRα overexpression enhanced cell proliferation and migration of SNB-19, SF-295, A172, T98G, LN229, and LN18 glioma cell lines in vitro and ERRα silencing reduced glioma growth in a xenograft model [221]. On the other hand, recently Wu et al., (2012) [222] investigated ECS levels and expression, in human low grade and high-grade glioma tissues showing that 2-AG levels increased in more malignant glioma (WHO grade III and IV) while CB1 and CB2 receptors expression levels were elevated in human glioma compared to normal tissues [222]. Based on these findings it is conceivable that ECS is related to glioma cancer progression and could cooperate with the estrogen system in controlling human glioma cell proliferation and transformation. Therefore, targeting simultaneously CB and ER receptors could be a therapeutic strategy in the management of brain cancers. Moreover, the fact that ECS and estrogen system activate and share several cell signaling pathways such as the MAPK, ERK 1/2, PI3K, and *C*–jun −*N* terminal kinase typically involved in cancer cell proliferation [9] empowers the importance of investigating the simultaneous modulation of these two systems in cancer to develop new pharmacologic approaches.

## 8. Closing Remarks

The interplay between ECS and estrogen is well documented in brain development and several physiological processes of the CNS like memory and learning or the central control of reproduction; the same occurs at the periphery, primarily in the gonads and reproductive tract. Particular interest deserves the estradiol dependent expression of FAAH, the key enzyme in the modulation of endocannabinoid tone and therefore a primary actor in driving endocannabinoid signaling. In this respect, environmental factors like chemicals that mimic estradiol signaling or impaired levels of endogenous estradiol may have deep consequences for health, particularly on reproduction and behavior, by means of direct (or indirect) modulation of endocannabinoid signaling. ECS plays a protective role in steroid hormone-dependent tumors, but not in brain cancer. However, ECS may represent a prognostic/diagnostic/therapeutic biomarker for cancer and the simultaneous investigation of ECS and ES system may result useful for the development of pharmacologic approaches, particularly in diseases that show a very different incidence and maybe different features of pathogenesis in both sexes.

## Figures and Tables

**Figure 1 ijms-22-00972-f001:**
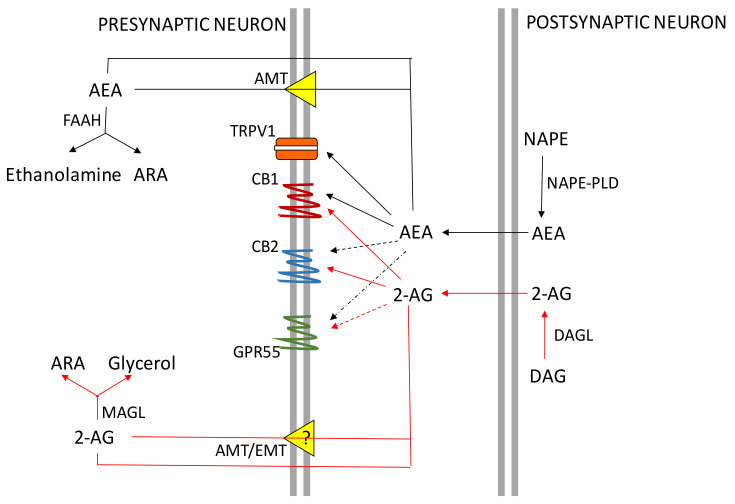
Schematic representation of the ECS in neurons. 2-AG: 2-arachidonoylglycerol; AEA: anandamide (N-arachidonoylethanolamide); ARA: Arachidonic acid; ATM: AEA membrane transporter; CB1: Type 1 cannabinoid receptor; CB2: Type 2 cannabinoid receptor; DAG: diacylglycerol; DAGL: DAG lipase ETM: endocannabinoid membrane transporter; FAAH; fatty acid amide hydrolase; GPR55: orphan G protein-coupled receptor 55; MAGL: monoacylglycerol lipase; NAPE: *N*-acylphosphatidylethanolamide; NAPE-PLD: NAPE-phospholipase D; TRPV1: transient receptor potential cation channel subfamily V member 1.

**Figure 2 ijms-22-00972-f002:**
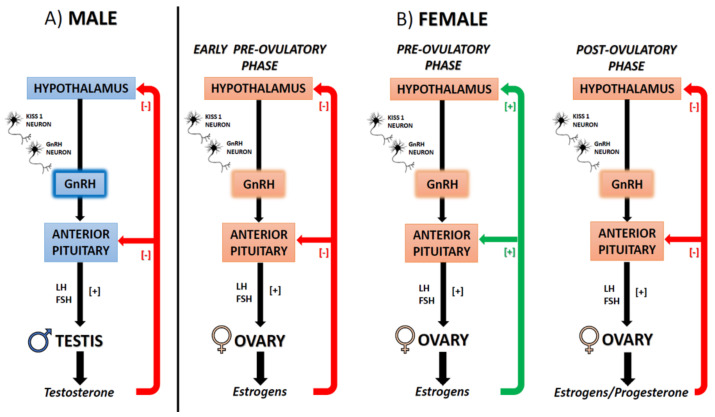
Schematic representation of the hypothalamus-pituitary gonad (HPG) axis in mammals. (**A**) Male (**B**) Female at different phases of the ovulatory cycle. Gonadotropin-releasing hormone (GnRH); Luteinizing hormone (LH); Follicle Stimulating Hormone (FSH).

**Figure 3 ijms-22-00972-f003:**
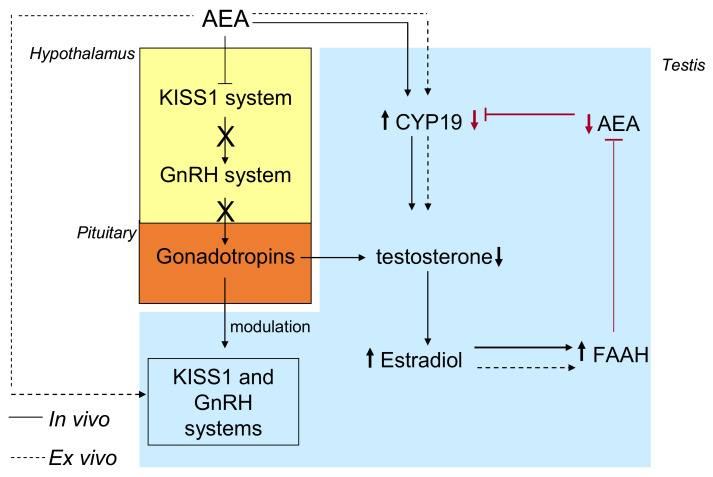
Centrally-mediated, and intratesticular mechanisms in the control of sex steroid biosynthesis through the modulation of FAAH in the frog. Black lines: centrally mediated effects of AEA; dotted black lines: intratesticular effects of AEA; red lines: the possible regulatory feedback loop on AEA tone.

**Table 1 ijms-22-00972-t001:** ECS modulation and its effects in hormone-dependent cancers.

Hormone-Dependent Cancer	Type of Study/Experimental Model	Main Results	Reference
Hepatocellular carcinoma (HCC)	HCC samples from patients at different stages of carcinogenesis	Overexpression of CB1 and CB2 receptor mRNA and protein expression levels.	[199]
Human prostate cancer (PCC)	PRL responsive DU-145 cells	AEA and 2-AG inhibited PRL—induced DU-145 cell proliferation.	[200]
	AR-positive (LNCaP and 22RV1) and negative (DU-145 and PC-3) cells. Mouse model of xenograft tumor.	AEA and Cannabidiol (CBD) inhibited PCC cell proliferation and potentiated the effects of bicalutamide and docetaxel against LNCaP and DU-145 xenograft tumors.	[201,202]
	Human PCC tissues and normal (healthy) prostate tissues	CB1 receptor and TRPV1 mRNAs and protein levels were higher in PCC. TRPV1 correlated with increasing PCC tumor grades, whilst CB1 receptor levels were not.	[203,204]
Breast cancer (BC)	MMTV-neu mouse model of ErbB2-driven metastatic breast cancer	Δ^9^-THC, marijuana and JWH-133 (CB2 receptor agonist) reduced cancer cell proliferation, impaired tumor angiogenesis, and reduced lung metastases.	[205]
	Human ErbB2 BC samples	Overexpression of CB2 receptor correlated with ErbB2 expression.	
	MCF-7, MDA-MB231 and EFM-19 and T47D human BC cells	CB1 receptor stimulation by AEA reduced adhesion and migration of BC cells and AEA and 2-AG inhibited the nerve growth factor (NGF)−; inhibition of PRLr levels via CB1 receptor modulation of PRL responsive BC cells. AEA inhibited epithelial-mesenchymal transition of BC cells.	[22,200,206,207]
Endometrial cancer (EC)	Hec50co and Ishikawa cells	eCBs and CBD reduced cell viability inducing apoptosis.	[208]
Colorectal cancer (CRC)	Azoxymethane (AOM)- and dextran sulfate sodium (DSS)-induced CRC mouse models. *Cnr1*^−/−^ and *Cnr1*^−/−^/*GPR55*^−/−^ double knockout mice	The putative GPR55 receptor acted as an oncogene and the CB1 receptor as a tumor suppressor.	[209]
	Caco2, DLD-1, and Sw620 CRC cell lines; AOM-induced CRC mouse model	The CB1 receptor antagonist/inverse agonist SR141716 inhibited cell growth and reduced precancerous lesions in the mouse colon. Increased AEA levels also reduced CRC cell proliferation. The inhibitory effect of AEA was reached by 17β estradiol driven up-regulation of the CB1 receptor.	[6,210]
	CRC patients	Hypermethylation of the *CNR1* gene in CRC patients and upregulation of CB1 receptor in normal tissues.	
	Colon biopsy by patients. CaCo-2 and DLD-1 cell lines	AEA and 2-AG levels were elevated in adenomas and CRCs. AEA and 2-AG treatment inhibited CRC cell proliferation in a CB1 and CB2 receptors dependent manner.	[211]

## Data Availability

Data sharing not applicable.

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
