# Peer review of "The Complex Interplay between Endocannabinoid System and the Estrogen System in Central Nervous System and Periphery"

_ijms, 2021, doi:10.3390/ijms22020972_

Round 1

Reviewer 1 Report

Please, find the detailed review in the provided file. The topic of the manuscript is interesting, but too many typical and stylistical errors were made. English need improvement before publication.

Author Response

We thank the reviewer for her/his constructive comments. We addressed all the queries and the resided version of the manuscript has been significantly improved. In details:

  • We have corrected several typos all over the text, following the annotation on the provided pdf file.
  • A figure on ECS has been added (figure 1) in par.2.
  • Information concerning nongenomic estrogen receptors has been added in par 3 (151-169) and in the context of endocannabinoid system (607-699).
  • A new paragraph entitled “ES and ECS modulation by estrogens-like substances affecting CNS and gonads” (par 6) comprising the revised old par 5.2 and the effects of estrogen like substances in the CNS has been added.

Reviewer 2 Report

This review is devoted to the problem of interaction of the endocannabinoid system and the estrogen system in the central nervous system and peripheral tissues. The topic is relevant, as it reveals the mechanisms of the cooperative effects of various signaling systems on physiological functions, and also reveals possible undesirable consequences of the use of narcotic and pharmacological substances.
The article collects and summarizes the current data in this area. The review is well structured, written in clear language, adequately cited, and reasonably well illustrated.
I would recommend publishing the article as it is.

Author Response

We thank the reviewer for her/his positive comments.

Reviewer 3 Report

Ref: ijms-1064182

Title: The complex interplay between Endocannabinoid System and the Estrogen System in Central Nervous System and periphery (Journal: IJMS)

Recommendation: Major Revision

Well written review. Nicely described and informative schemes are an additional value. The topic of the review – interplay between endocannabinoid system and estrogen system in CNS will certainly find many readers.

Comments:

  1. Nothing is mentioned about nongenomic estrogen receptors e.g. GPER1, mERα, mERβ, ER-X, GqmER in the context of endocannabinoid system.
  2. Since both ECS and ERs interact with PPAR receptors perhaps it would be interesting to describe this.

Author Response

  • Nothing is mentioned about nongenomic estrogen receptors e.g. GPER1, mERα, mERβ, ER-X, GqmER in the context of endocannabinoid system.

Information concerning nongenomic estrogen receptors has been added all over the text and in the context of endocannabinoid system.

  • Since both ECS and ERs interact with PPAR receptors perhaps it would be interesting to describe this.

The possible interplay among ECS, estrogens and PPAR has been discussed in  a new paragraph entitled “ES and ECS modulation by estrogens-like substances affecting CNS and gonads” (par 6). This paragraph comprises the revised old par. 5.2 and additional data concerning the effects of estrogen like substances in the CNS.

Round 2

Reviewer 3 Report

In my opinion, the review is ready for publication.